# Recognition of Vehicles Entering Expressway Service Areas and Estimation of Dwell Time Using ETC Data

**DOI:** 10.3390/e24091208

**Published:** 2022-08-29

**Authors:** Qiqin Cai, Dingrong Yi, Fumin Zou, Zhaoyi Zhou, Nan Li, Feng Guo

**Affiliations:** 1School of Mechanical Engineering and Automation, Huaqiao University, Xiamen 361021, China; 2Fujian Key Laboratory for Automotive Electronics and Electric Drive, Fujian University of Technology, Fuzhou 350118, China; 3Digital Fujian Traffic Big Data Research Institute, Fujian University of Technology, Fuzhou 350118, China; 4College of Computer and Data Science, Fuzhou University, Fuzhou 350108, China

**Keywords:** VR-XGBoost, K-VDTE, ETC data, ESAs, data mining

## Abstract

To scientifically and effectively evaluate the service capacity of expressway service areas (ESAs) and improve the management level of ESAs, we propose a method for the recognition of vehicles entering ESAs (VeESAs) and estimation of vehicle dwell times using electronic toll collection (ETC) data. First, the ETC data and their advantages are described in detail, and then the cleaning rules are designed according to the characteristics of the ETC data. Second, we established feature engineering according to the characteristics of VeESA and proposed the XGBoost-based VeESA recognition (VR-XGBoost) model. Studied the driving rules in depth, we constructed a kinematics-based vehicle dwell time estimation (K-VDTE) model. The field validation in Part A/B of Yangli ESA using real ETC transaction data demonstrates that the effectiveness of our proposal outperforms the current state-of-the-art. Specifically, in Part A and Part B, the recognition accuracies of VR-XGBoost are 95.9% and 97.4%, respectively, the mean absolute errors (*MAE*s) of dwell time are 52 and 14 s, respectively, and the root mean square errors (*RMSE*s) are 69 and 22 s, respectively. In addition, the confidence level of controlling the *MAE* of dwell time within 2 min is more than 97%. This work can effectively recognize the VeESA and accurately estimate the dwell time, which can provide a reference idea and theoretical basis for the service capacity evaluation and layout optimization of the ESA.

## 1. Introduction

By the end of 2021, the total mileage of expressways in China was approximately 169,100 km, ranking first in the world [1]. As an essential and critical core node of expressways, ESA is of great significance in regulating road traffic flow and relieve traffic pressure. However, the infrastructure of most ESAs built in the early years has been unable to meet the demand of the rising traffic volume, resulting in frequent queues and congestion [2,3]. Therefore, a scientific and reasonable evaluation of the service capacity of ESA and further quantitative suggestions for the reconstruction and extension of ESA have become urgent issues at present [4,5]. The pause rate and dwell time of vehicles are essential parameters in the operation and management of ESA. It is not only an important metric for operation evaluation but also a premise for layout optimization. Therefore, it is of great practical significance and application value to accurately estimate the pause rate and dwell time for the quantification of the reconstruction and extension of ESA.

Currently, the main methods for pause rate estimation are the elastic coefficient method [6,7] and feature engineering method [8,9,10,11,12,13,14,15,16,17,18]. The elasticity coefficient method, proposed by the Japanese expressway design standards, mainly uses ESA pause rate survey data and national economic data to establish the pause rate trend model to estimate the future pause rate. Although this method is simple in principle and convenient in calculation, it has certain limitations and one-sidedness. On the other hand, the feature engineering method mainly considers multidimensional features such as major road traffic flow, average speed and human physiological demand and feeds the model for training and learning to estimate the pause rate. However, few studies have investigated the estimation of dwell time in ESA. The limited survey data obtained at a specific ESA are mainly used in existing studies to statistically analyze the vehicle dwell time according to categories such as vehicle type, diurnal differences, seasonal differences, etc. [19,20,21,22,23,24]. Therefore, the following challenges remain for pause rate and dwell time estimation. (1) Fewer and more difficult to obtain ESA data, resulting in model training effects that could be further improved. (2) Existing studies tend to consider only the overall estimation of the pause rate, ignoring the differences between individual vehicles. (3) The dwell time estimation fails to fully consider the ESA regionality, timeliness and kinematic principle in the vehicle travel law.

To address the aforementioned challenges, we propose a method for the recognition of vehicles entering ESAs (VeESAs) and the estimation of dwell times using ETC data. First, with the rapid development of Internet of Vehicles (IoV) technology in recent years [25,26], China built the world’s largest IoV system—the ETC system—at the end of 2019, with a penetration rate of more than 80% of its users. Therefore, this study will utilize ETC data as experimental data to solve the problem of insufficient data. Then, ETC data preprocessing rules are designed by deeply mining the characteristics of ETC data. Second, we proposed an XGBoost-based VeESA recognition (VR-XGBoost) model based on a detailed analysis of the main factors affecting VeESA. On this basis, taking into full consideration the driving pattern of vehicles entering/exiting the ESA, we proposed a kinematics-based vehicle dwell time estimation (K-VDTE) method, which is expected to provide reference ideas for the scientific and reasonable evaluation of the service capacity of the ESA. This work can provide decision support for the layout optimization of ESA reconstruction and extension and improve the management level and high-quality development of ESA.

The main contributions of this study are as follows:We proposed a VR-XGBoost model for recognizing vehicles entering expressway service areas based on ETC data, which not only achieves an effective estimation of the pause rate but also accurately recognizes individual vehicles driving into ESA.Taking into full consideration the driving pattern of vehicles entering/exiting the ESA, we proposed a K-VDTE model for vehicle dwell time estimation.The validity of the proposed method is verified by using real ETC data, which can provide a more scientific and reasonable reference basis for ESA reconstruction and extension.

The remainder of this work is organized as follows: Section 2 reviews related work regarding ESA pause rate and vehicle dwell time estimation. The proposed method, including the framework, data preprocessing, feature engineering, the VR-XGBoost model, and the K-VDTE model, is described in Section 3. Section 4 shows the experimental results and analysis. Finally, the conclusion is presented in Section 5.

## 2. Related Work

### 2.1. Pause Rate Estimation

In this section, an overview of pause rate estimation methods is presented. The elastic coefficient method (ECM) was proposed in early Japanese expressway design standards for calculating the pause rate of various types of VeESAs [6]. Drawing on relevant experience in Japan, Sun et al. [7] concluded that ECM was also applicable to the development pattern of the ESA pause rate in Guangdong Province, China, and used the ECM to estimate the average growth rate of the pause rate to achieve prediction.

Considering the close relationship between the pause rate and ESA spacing, Cui et al. [8] proposed a new method for determining the pause rate based on the continuous vehicle travel time. Through an in-depth analysis of the relationship between the pause rate and traffic flow parameters [9,10], Chen et al. [11] proposed a pause rate estimation method based on a traditional linear regression model, which provided an important reference basis for the layout optimization and function design of ESA. In response to the low accuracy of pause rate prediction, a BP neural network-based ESA pause rate prediction model was constructed [12]. On the basis of previous work, Shen et al. [13] extracted multidimensional feature vectors from the data and constructed a tree-level BP neural network for pause rate prediction, which further improved the prediction accuracy. To further optimize the essential parameters of the wavelet neural network (WNN), some scholars introduced evolutionary algorithms, such as particle swarm optimization (PSO) [14] and genetic algorithm (GA) [15], to optimize the initial parameters of the WNN. The improved WNN-based pause rate prediction models were established, and the validity and reliability were verified on a real dataset. Under the premise of fully investigating the global optimal search capability of particles, Sun et al. [16] improved the topology of traditional PSO and fused it with the XGBoost algorithm to form a combined model for ESA traffic flow prediction. Experiments have demonstrated that the combined model has higher prediction accuracy and stronger generalization ability than a single model.

In the past few decades, deep learning methods [27,28], such as long short-term memory (LSTM) and convolutional neural networks (CNN), have achieved good performance in the field of transportation and are widely used in traffic flow prediction. Wang et al. [17] built a model based on LSTM for ESA instantaneous population analysis and prediction. The experimental results showed that it was able to accurately predict population mobility despite the relatively large population fluctuations. Zhao et al. [18] extracted spatiotemporal features using CNN, LSTM, and attention mechanism models and proposed a short-term traffic flow prediction model based on STL-OMS to achieve an accurate prediction of ESA traffic flow.

### 2.2. Vehicle Dwell Time Estimation

In this section, an overview of dwell time estimation methods is presented. King et al. [19] conducted an early field survey at nine locations in the United States. The results showed that the average vehicle dwell time in rest areas was 11.4 min, with a standard deviation of 12.87 min, a minimum dwell time of 1 min and a maximum dwell time of 3 h and 31 min. Recently, the Japanese Institute of Expressway General Technology noted through actual statistics that the average dwell time of small vehicles in most ESAs exceeded 25 min [20], while the dwell time for families with elderly and children was extended by an average of 10~20 min in ESAs [21]. Furthermore, analysis of dwell time by vehicle type showed that heavy vehicles had the longest average dwell time, significantly longer than other vehicle types [22]. Analysis of dwell time by seasonal differences showed that all categories of vehicles had longer dwell times in summer than in any other season [23]. Analysis of dwell time by diurnal differences showed that the average dwell time was significantly longer at night than during the day [24].

In addition, Hirai et al. [29] estimated the total dwell time in the service area for the whole trip by mining the ETC trip data using the average travel speed method. The correlation analysis of the dwell time distribution characteristics and rest behavior [30] was expected to construct the next rest behavior model. At the same time, the driver’s rest behavior was used to characterize the distribution of vehicle travel time [31] to further construct driving behavior characteristics [32]. A method for calculating the number of stranded vehicles across time was proposed through statistical analysis of vehicle dwell time and rest behavior characteristics [33], and then a mathematical model for ESA scale design was proposed [34], which was used to optimize the ESA layout [35,36].

## 3. Methodology

### 3.1. Framework

In this section, we present the framework of this study, as shown in Figure 1. First, we perform data preprocessing, including extraction of required data, ETC trajectory construction, data cleaning, data fusion and forming of structured data. Second, we consider features such as speed features, spatiotemporal features and external features to construct feature engineering, thus building an XGBoost-based VeESA recognition model. On this basis, a kinematics-based vehicle dwell time estimation model is proposed. This study not only enables the effective recognition of VeESA but also further estimates their dwell time in the service area.

### 3.2. Data Overview and Preprocessing

#### 3.2.1. Data Overview

The experimental datasets in this work contain the ETC dataset and ESA dataset. The ETC data were collected by more than 1000 ETC gantries deployed in the whole road network of the Fujian Provincial Expressway. Specifically, as the world’s largest IoV system, the ETC system uses radio frequency identification (RFID) technology to enable mobile vehicles equipped with an onboard unit (OBU) to communicate with roadside units (RSU) for data collection [37]. The collection period was from 3 to 10 September 2020. We obtained a total of 42,964,489 ETC data, including vehicle ID (after desensitization), transaction time, gantry ID, vehicle type, etc., as shown in Table 1. According to the classification of vehicle types and tolls of China’s expressway, vehicles can be divided into 4 categories of buses, 6 categories of trucks and 6 categories of special operating vehicles. The total number of vehicles is approximately 1.72 million in the dataset. Specifically, each transaction data contains all field information.

The ESA data were collected by the cameras at the entrance and exit of Yangli ESA Part A/B. Specifically, the camera uses the technology of license plate recognition to obtain information about the vehicles entering the service area [38]. The collection period is consistent with the ETC data. We obtained more than 30,000 data points, including vehicle ID (after desensitization), capture time, service area ID and entrance/exit information, as shown in Table 2. The total number of vehicles is approximately 18,000 in the dataset. It is worth noting that this dataset is only used for experimental validation to evaluate the recognition effect and the estimation accuracy of vehicle dwell time.

In this work, only the data of Yangli ESA and two ETC gantries before and after it are used, whose deployment locations are shown in Figure 2. To facilitate the following explanation, we have made relevant definitions as follows.

**Expressway Section *QD*** [39]: Each ETC gantry and the entrance/exit of an expressway toll station is collectively called a node G, and two adjacent nodes constitute an expressway section, referred to as ***QD***:(1)$$QD=\langle G_{1},\;G_{2}\rangle \;$$
where $G_{1}$ and $G_{2}$ are the start and end points of ***QD***.

Taking road upline as an example, it can be seen from Figure 2 that $G_{1}$ and $G_{2}$ constitute **Section** 1 ($QD_{1}$), $G_{2}$ and $G_{3}$ constitute **Section** 2 ($QD_{2}$), where the ESA is located, and $G_{3}$ and $G_{4}$ constitute **Section** 3 ($QD_{3}$). It can be found from the partial enlarged detail that the gantries always appear in pairs, which are distributed along the upline and downline of the road, such as $G_{2}$ and $G_{2}^{\prime }$. Therefore, the discrete ETC data need to be processed into vehicle trajectories and fused with the ESA data, as detailed in Section 3.2.2.

#### 3.2.2. Data Preprocessing

The prerequisite for effective data mining is to ensure data quality. However, there is a large amount of “dirty” data in ETC data, which is caused by various objective factors such as equipment failure, wireless signal crosstalk and bad weather in the process of ETC data collection, transmission and storage, which seriously affects the potential value of ETC data mining. There are 3 main problems in the “dirty” data as follows:(1)Data Redundancy

Generally, it is generated by repeated uploading of data in the transmission process or repeated copying in the storage process. This tends to cause an increase in the data scale and serious interference with data mining. In addition, the continuous communication between vehicle OBUs and ETC antennas due to traffic congestion and anchor failure within the antenna coverage area is also a cause of data redundancy. In general, it is sufficient to keep only one of the instances of data and delete the rest directly.

(2)Data Missing

Due to equipment failure, bad weather and other reasons, the vehicle OBU does not communicate or communicates unsuccessfully with the ETC antenna, which results in missing data. At the same time, there is also the possibility of missing data due to network packet loss during data transmission.

(3)Data Abnormality

With the influence of wireless signal crosstalk and other factors, the vehicle OBU of the vehicle traveling on the road upline communicates successfully with the ETC antenna deployed on the road downline, and the dataset generates records that do not comply with expressway driving rules.

However, due to the highly discrete characteristic of ETC data, it is difficult to achieve effective judgment of data abnormalities due to isolated data points. Therefore, it is necessary to rely on the trajectory semantic context formed by the topology of the expressway ETC gantry network to accurately detect and repair the above situation. For this purpose, we further define it as follows: **ETC Trajectory**
eTr: The sequence of ETC gantry nodes formed by a vehicle passing through a continuous expressway **Section** $\langle QD_{1},QD_{2},\ldots ,QD_{n-1}\rangle$ is called an ETC trajectory eTr:(2)$$eTr=\langle tr_{1},tr_{2},\ldots ,tr_{n}\rangle$$
where $tr_{1}$ and $tr_{n}$ are the start and end points of the trajectory, respectively. $tr_{i}$ is the transaction data when the vehicle travels through the ETC gantry, which contains information such as gantry ID $tr_{i.N}$, transaction timestamp $tr_{i.T}$, vehicle ID $tr_{i.P}$, vehicle type $tr_{i.C}$, entrance gross axle weight $tr_{i.EW}$, entrance ID $tr_{i.EID}$, entrance timestamp $tr_{i.ET}$ and workday $tr_{i.H}$ (consistent with Table 1). *n* indicates the total number of nodes that the vehicle passes through.

The ETC data cleaning algorithm (Algorithm 1) includes the construction of the vehicle trajectory, data cleaning and data repair. First, the ETC data are grouped by vehicle ID  tri.P, entrance ID tri.EID, and entrance timestamp tri.ET. Second, we eliminate duplicate data after sorting by transaction timestamp tri.T for each set of data. Third, we obtain two adjacent data in each set of data and judge the correctness by its topological information, which mainly includes the removal of redundant data generated by the opposite gantries and the repair of missing data. It is worth noting that the topology dataset includes two subsets: TP and TP′, which is a collection of topologies (e.g., 〈G1,G2〉). Specifically, TP={〈G1,G2〉,〈G2,G3〉,〈G3,G4〉,…} denotes normal topology data and TP′={〈G1,G2′〉,〈G2,G3′〉,〈G3,G4′〉,…} denotes opposite topology data. The topologies in TP and TP′ always appear in pairs, such as 〈G1,G2〉 and 〈G1,G2′〉. Finally, the vehicle trajectories that meet the requirements are added to the trajectory dataset eTRAJ. The specific algorithm is shown as follows:
**Algorithm 1****:** ETC data cleaning algorithm**Input:** ETC data **eData**, Topology data ***TP***, Opposite topology data ***TP′*****Output:** ETC trajectory dataset eTRAJ1: G_eData = **eData**.Groupby([trp,trEID,trET]); # Grouping2: **For**
eTrj∈ G_eData **do**: # Traversal operation for each set of data3:        eTrj←eTrj.sorted(by=trT) # Sorted by transaction time4:        eTrj←eTrj.drop_duplicates()# Data deduplication5:        While (*i*=1, *i* < len(eTrj)):6:                   tp←〈tri.N,tri+1.N〉7:                   **IF**
tp∈TP**:**8:                               i+=1;9:                               continue;10:                   **Else IF**
tp∈TP′:11:                            tp′←〈tri.N,tri+2.N〉12:                            IF:
tp′∈TP13:                                        delete tri+1# Delete opposite gantry transaction data14:                                        *i*+= 2;15:                            Else:16:                                        tp′′←〈tri.N,tri+1.N〉, tp′′′←〈tri+1.N,tri+2.N〉17:                                     **IF**
tp′′∈TP**&&**
tp′′′∈TP:18:                                                       tri+1.N←tri+1.N′# Replacement of opposite gantry ID19:                                                       *i*+= 2;20:                                     Else:21:                                              break;22:                                     End IF23:                               End IF24:                      Else:25:                               break;26:                      End IF27:                      IF *i* = len(eTrj)-1:28:                               eTRAJ.append(eTrj);29:                               End IF30:       End While31: End For

The ETC driving trajectory through data cleaning also needs to be fused and matched with the service area traffic data as the label data for subsequent experiments. Therefore, we designed algorithm for fusion of ETC trajectory and ESA data (Algorithm 2). As seen from Section 3.2.1, the ESA is located in $QD_{2}$. Therefore, only the ETC driving trajectory data and service area data vehicle data must be obtained, and at the same time, the service area entrance and exit capture time in the 2nd and 3rd gantry transaction time periods can match the VeESA to the corresponding ETC driving trajectory. The remaining unmatched driving trajectories are not driven into the service area trajectories.

Notably, the gantry system and the service area entrance/exit camera system appear to be clocked out of sync. Therefore, the time difference delta is set. We make the transaction time of G2 Δt hours ahead and the transaction time of G3 Δt hours behind, i.e., tr2.Tj−Δt and tr3.Tj+Δt. By expanding the time range, we ensure that VeESA is fully matched. After the experiments, the time difference in this work is set to 1 h, i.e., Δt=1 h. The specific algorithm is shown as follows:
**Algorithm 2:** Fusion of ETC trajectory and ESA data**Input:** ETC trajectory dataset ***eTRAJ***, ESA dataset ***sData***, time difference ∆*t***Output:** final trajectory data eTr1: VIDSet=unique(sData. VehID)2: **For** eTrj∈eTRAJ
**do**:3:       eTrj=〈trPj,tr1,Tj,tr2,Tj,tr3,Tj,tr4,Tj,trCj,trEIDj,trETj,trWj,trHj〉;4:       trlj←0; trPCTj←null; trNCTj←null;5:       **If** trPj **in** VIDSet:6:               sdTmp=sData[sData. VehID==trPj]7:                **For** row **in** sdTmp.iterrows():8:                         **IF**
tr2.Tj−Δt< row.CapTime *<*
tr3.Tj+Δt:9:                                    trlj←1;10:                                 **IF** row.ExEn = 0:11:                                        trPCTj←row.CapTime;12:                                **Else**:13:                                        trNCTj←row.CapTime;14:                               **End IF**15:                         **Else**:16:                           continue;17:                         **End IF**18:             **End For**19:       **Else:**20:             continue;21:       **End IF**22:       eTrj.append(〈trPCTj,trNCTj,trlj〉)23: End For

Through data cleaning and data fusion, a total of approximately 44,000 and 39,000 trajectories were obtained in Yangli Part A and Part B, respectively. The final data samples are shown in Table 3. In these trajectories, the total ETC trajectories of entering Part A and Part B are approximately 7800 and 6700, respectively, and the pause rates of both Parts A and B are approximately 17%. It is worth noting that due to equipment failure and other reasons, there is a missing situation of service area entrance/exit capture data in the experimental dataset. However, this problem does not affect the experiments on the recognition of VeESA in this work. In other words, only one valid capture of data needs to exist in the ESA entrance/exit data to complete the tagging work. Subsequent vehicle dwell time estimation experiments will be conducted by selecting the trajectories where both entrance and exit capture data exist.

### 3.3. XGBoost-Based VeESA Recognition

#### 3.3.1. Feature Vector Modeling

There are numerous factors that affect the pause rate and dwell time of ESA, which have highly nonlinear characteristics. Therefore, we summarize the previous research results [15,18] and construct feature vectors from 3 dimensions, such as speed features, spatiotemporal features, and external factors. The details are as follows:(1)Speed Features

The speed features are the key features for the recognition of VeESA. When a vehicle enters the ESA, the average speed of ESA section $\left(QD_{2}\right)$ will be significantly lower than $QD_{1}$ and $QD_{2}$. Meanwhile, it will also be lower than the overall average speed of other vehicles of the same type in this section. Therefore, we construct the speed feature vector as follows:(3)$$v={\left(v_{1},v_{2},v_{3},v_{4}\right)}^{T}$$
where $v_{1}\sim v_{3}$ represent the driving state of the individual vehicle during the whole trip. Among them, $v_{1}=d_{1}/\left(tr_{2.T}-tr_{1.T}\right)$ indicates the average speed of the vehicle in $QD_{1}$, $v_{2}=d_{2}/\left(tr_{3.T}-tr_{2.T}\right)$ represents the average speed of the vehicle in $QD_{2}$, $v_{3}=d_{3}/\left(tr_{4.T}-tr_{3.T}\right)$ represents the average speed of the vehicle in $QD_{3}$, represent the total mileage of $QD_{1},QD_{2}$ and $QD_{3}$, respectively, and $v_{4}=\frac{1}{n}\overset{n}{\underset{j=1}{\sum }}v_{2}^{j}$ represents the overall average speed of vehicles of the same type, except for the vehicle in $QD_{3}$. $v_{4}$ mainly avoids the disturbance caused by the reduction in $v_{2}$ in certain time periods due to special conditions or traffic congestion.

(2)Spatiotemporal Features

In general, the longer a vehicle spends on the expressway, the more demands on the ESA for drivers and passengers. Therefore, we construct the actual cumulative travel time of the vehicle from the entrance of the toll station to the ESA as one of the spatiotemporal features. At the same time, people’s needs for ESA are also different during different times of the day and on non-workdays. For example, the pause rate of ESA is generally higher at meal times, after midnight and on non-workdays. Therefore, we construct the spatio-temporal features vector as shown below.
(4)$$\gamma ={\left(\gamma_{1},\gamma_{2},\gamma_{3}\right)}^{T}$$
where $\gamma_{1}$ represents the actual cumulative travel time of the vehicle from the entrance of the toll station to the ESA, $\gamma_{2}$ represents the time period feature, which divides the whole day into 24 time periods by hour, whose value range is 0~23, and $\gamma_{3}$ is a variable for the workday, and its value is 0 (workday) or 1 (non-workday).

(3)External Features

Vehicle type is also an important feature in road traffic. Different types of vehicles have different demands on the ESA. At the same time, the difference in passenger/freight volume will also have some influence on the pause rate of ESA. For example, the more passengers a bus carries, the more stops it needs for rest, dining, etc. Fully loaded large trucks often require services such as breaks and water refills. Therefore, the feature vector is constructed as follows:(5)$$\theta ={\left(\theta_{1},\theta_{2},\theta_{3}\right)}^{T}$$
where $\theta_{1}$ represents vehicle type. From the data source of Section 3.2.1, the vehicle types are divided into 16 categories, $\theta_{2}$ represents passenger/freight volume, and $\theta_{3}$ represents the traffic flow of the same time slice.

Feature vector modeling is completed by constructing all feature values into vector form.

#### 3.3.2. Modeling of Recognition of VeESA

XGBoost is an integrated learning method based on the boosting algorithm, whose learner usually chooses the decision tree model [40], as shown in Figure 3. The model learns the residuals of the true values and the predicted values of the decision tree by iteratively generating new decision trees. Eventually, the results of all trees are accumulated as the final result to obtain better classification accuracy, i.e., the weak classifiers are combined into a stronger classifier. Therefore, we introduce XGBoost to build a VeESA recognition model.

We abstracted a 10-dimensional feature vector from the raw ETC data with known label information to form the sample dataset. We set the dataset as $S=\left\{\left(x_{1},y_{1}\right),\left(x_{2},y_{2}\right),\ldots ,\left(x_{n},y_{n}\right)\right\},$ where $x_{i}={\left(v_{1},v_{2},v_{3},v_{4},\gamma_{1},\gamma_{2},\gamma_{3},\theta_{1},\theta_{2},\theta_{3}\right)}^{T}\left(i=1,2,\ldots ,N\right)$ represents the feature vector of the *i*-th sample. $y_{i}=0/1\left(i=1,2,\ldots ,N\right)$ represents the classification label value corresponding to $x_{i}$. We assume that VR-XGBoost integrates K decision trees, and the prediction result is shown in Equation (6):(6)y^i=∑k=1Kfk(xi) , fk∈F
where *K* represents the number of trees, $f_{k}\left(x_{i}\right)$ represents the predicted value of the *k*-th decision tree on sample $x_{i}$, and *F* represents the integrated classifier composed of all decision trees.

The objective function of XGBoost consists of the loss function and the regularization item, as shown in Equation (7):(7)Obj=∑i=1nloss(yi,y^i)+∑k=1KΩ(fk)
where loss represents the logistic regression loss function used for classification.
(8)loss(yi,y^i)=yiln(1+e−y^i)+(1−yi)ln(1+ey^i)  

$\Omega \left(f_{k}\right)$ represents the L1 regularizer, which is used to prevent the model from overfitting. The formula for the regularizer is Equation (9):(9)$$\Omega \left(f_{k}\right)=\alpha T_{k}+\frac{1}{2}\alpha ∥w_{k}{∥}_{1}$$
where α represents the regularization penalty coefficient, which takes values in the range of [0, 1]. $T_{k}$ presents the number of leaves of the k-th tree and $w_{k}$ represents the leaf weight of the k-th tree.

The XGBoost algorithm adopts an additive stepwise integration strategy in the training process. Tree-1 is optimized first, followed by Tree-2 until Tree-K has been optimized.
(10)y^i(0)=0
(11)y^i(1)=f1(xi)=y^i(0)+f1(xi) 
(12)y^i(2)=f1(xi)+f2(xi)=y^i(1)+f2(xi)…
(13)y^i(k)=y^i(k−1)+fk(xi)

We improve the prediction accuracy by adding an incremental function $f_{k}$ to optimize the objective function during the iterative process, which is calculated as in Equation (14):(14)Obj(k)=∑i=1nloss(yi,y^i(k−1)+fk(xi))+Ω(fk)+c
where c represents the constant term and y^i(k−1) denotes the predicted value in the k−1st iteration on sample $x_{i}$.

Next, we expand the second-order Taylor formula and discard the constant term to speed up the solution and reduce the running time, which is calculated as Equation (15):(15)Obj(k)=∑i=1n[l(yi,y^i(k−1))+gifk(xi)+12hifk2(xi)]+Ω(fk)=∑j=1K[(∑i∈Ijgi)wj+12(∑i∈Ijhi+αwj2)]+αT
where $I_{j}=\left\{i|q\left(x_{i}\right)=j\right\}$ denotes the sample set of leaf j, and $g_{i}$ and $h_{i}$ are the first derivative and the second derivative of the loss function, respectively.

The objective function is transformed into a quadratic $Obj^{\left(k\right)}$ minimization problem on $w_{j}$. Then, we obtain the optimal prediction of each leaf node and the minimum value of the objective function, that is, the optimal value:(16)$$w_{j}{}^{*}=-\frac{G_{j}}{H_{j}+\alpha }$$
(17)$${\left(Obj^{\left(k\right)}\right)}^{*}=-\frac{1}{2}\overset{T}{\underset{j=1}{\sum }}\frac{G_{j}{}^{2}}{H_{j}+\alpha }+\alpha T$$
where $G_{j}=\underset{i\in I_{j}}{\sum }g_{i},H_{j}=\underset{i\in I_{j}}{\sum }h_{i.}$

### 3.4. Kinematics-Based Dwell Time Estimation

The vehicle dwell time is also an essential parameter in the operation and management of ESA. Therefore, after the recognition of VeESA, we need to further estimate the dwell time in the ESA. From the location of the service area between Gantry 2 and Gantry 3, we know that the total travel time of the section consists of the actual travel time of the vehicle in the section and the dwell time. Therefore, the dwell time $\Delta t_{s}$ can be obtained as follows:(18)$$\Delta t_{s}=\Delta t_{QD2}-\Delta t_{r}$$
where $\Delta t_{QD2}=tr_{3.T}-tr_{2.T}$, $\Delta t_{r}$ represents the actual travel time, which is an unknown parameter.

Therefore, the vehicle dwell time estimation is converted into the actual vehicle travel time estimation. Since the traffic conditions of the expressway are relatively smooth, the expressway can approximate the free-flow state in noncongested and nonemergency conditions. Vehicles usually travel smoothly on the highway, so the average speed of $QD_{1}$ and $QD_{3}$ can be used as the speed of $QD_{2}$, and thus the actual travel time of $QD_{2}$ can be estimated:(19)$$\Delta t_{r}=\frac{d_{2}}{\frac{v_{1}+v_{2}}{2}}=\frac{2d_{2}}{v_{1}+v_{2}}$$

By substituting Equation (19) into Equation (20), we can obtain the following:(20)$$\Delta t_{s}=tr_{3.T}-tr_{2.T}-\frac{2d_{2}}{v_{1}+v_{2}}$$

Although the average speed method is simple and straightforward, it does not take into account the kinematics of the VeESA during the entrance/exit ramp. In general, VeESA goes through a total of five kinematic stages, including smooth driving upstream, decelerating into the ESA, dwelling in the service area, accelerating out of the ESA and smooth driving downstream, as shown in Figure 4.

Therefore, we construct a kinematics-based model for estimating the dwell time, where the actual travel time $\Delta t_{r}$ is redefined as follows:(21)$$\Delta t_{r}=\Delta t_{1}+\Delta t_{2}+\Delta t_{4}+\Delta t_{5}$$
where $\Delta t_{1}\sim \Delta t_{5}$ correspond to the time spent in each of the above five stages.Stage 1: smooth driving upstream

According to the principle of inertia, the driving state of this stage can be considered as the continuation of the previous section ($QD_{1}$). Therefore, we approximate Stage 1 as uniform motion. We take the average travel speed of $QD_{1}$ as the travel speed of Stage 1, and we can obtain the time spent in Stage 1:(22)$$\Delta t_{1}=\frac{\Delta s_{1}}{v_{1}}$$
where $\Delta s_{1}$ is the distance from Gantry 2 to the diversion point of the entrance ramp and $v_{1}$ is the average travel speed of $QD_{1}$.Stage 2: decelerating into the ESA

To a certain extent, the ramps in ESA are similar to the ramps at the entrance and exit of the expressway toll station. However, the ramps at the entrances and exits of toll stations are usually designed with large curvature, while the ramps at service areas are generally of small curvature or even similar to straight lines. This makes the vehicle smoother when driving in/out of the service area. Therefore, we approximate Stage 2, i.e., the deceleration driving process into the service area entrance ramp, as uniform deceleration linear motion. From stage 1, the initial velocity of uniformly decelerating linear motion is $v_{1}$. Let the velocity at the moment $\Delta t_{2}$ be $v_{\Delta t_{2}}$, the displacement be $\Delta s_{2}$ and the acceleration be $a^{-}$, which gives the following.
(23)$$v_{\Delta t_{2}}=v_{1}+a^{-}\Delta t_{2}$$
(24)$$v_{\Delta t_{2}}^{2}-v_{1}^{2}=2a^{-}\Delta s_{2}$$

We combine Equations (23) and (24) to obtain the following.
(25)$$\Delta t_{2}=\frac{2\Delta s_{2}}{v_{1}+v_{\Delta t_{2}}}$$
where $\Delta s_{2}$ is the distance from the diversion point of the entrance ramp to the service area.Stage 4: accelerating out of the ESA

Similarly, we approximate Stage 4, i.e., the service area exit ramp acceleration process, as uniformly accelerated rectilinear motion until the driving speed reaches a steady state. From Stage 4, it can be seen that the vehicle reaches a smooth state after moment $\Delta t_{4}$, whose driving speed is $v_{3}$. Meanwhile, we assume the initial velocity $v_{30}$ and acceleration $a^{+}$ of uniformly accelerated linear motion. From Equation (23), we can obtain the time spent in Stage 4:(26)$$\Delta t_{4}=\frac{v_{3}-v_{30}}{a^{+}}$$

In general [41], $a^{+}=0.8\sim 1.2\;m\cdot s^{-2}$.Stage 5: smooth driving downstream

This stage is similar to the smooth driving upstream. Therefore, we approximate Stage 5 as uniform motion. The driving state of the next section ($QD_{3}$) can be considered a continuation of Stage 5. We take the average travel speed of $QD_{3}$ as the travel speed of Stage 5, and we can obtain the time spent in Stage 5 as follows:(27)$$\Delta t_{5}=\frac{\Delta s_{5}}{v_{3}}$$
where $v_{3}$ is the average travel speed in the back section of the service area, and $\Delta s_{5}$ is the distance from the smooth point in Stage 4 to Gantry 3, which is expressed as follows:(28)$$\Delta s_{5}=v_{30}\Delta t_{4}+\frac{1}{2}a^{+}\Delta t_{4}{}^{2}$$

We substitute Equations (22) and (25)–(27) into Equation (21) to obtain the following.
(29)$$\Delta t_{r}=\frac{\Delta s_{1}}{v_{1}}+\frac{2\Delta s_{2}}{v_{1}+v_{\Delta t_{2}}}+\frac{v_{3}-v_{30}}{a^{+}}+\frac{\Delta s_{5}}{v_{3}}$$

After finishing, we obtain the mathematical model for the estimation of vehicle dwell time based on kinematics.
(30)$$\Delta t_{s}=\Delta t_{QD2}-\left(\frac{\Delta s_{1}}{v_{1}}+\frac{2\Delta s_{2}}{v_{1}+v_{\Delta t_{2}}}+\frac{v_{3}-v_{30}}{a^{+}}+\frac{\Delta s_{5}}{v_{3}}\right)$$

It can be generally considered that the velocity $v_{\Delta t_{2}}$ in uniformly decelerating linear motion and the initial velocity $v_{30}$ in uniformly accelerating linear motion are both zero, which can be simplified as follows:(31)$$\Delta t_{s}=\Delta t_{QD2}-\frac{\Delta s_{1}+2\Delta s_{2}}{v_{1}}-\frac{v_{3}\left(v_{3}-2\right)}{2a^{+}}$$

## 4. Experiments

The experimental platform is a Centos Linux release 7.9.2009 (Core) operating system based on an Intel(R) Core (TM) i9-10900K CPU @ 3.70 GHz and 64 GB RAM, and all experiments were implemented on the open-source web application Jupyter Notebook using Python version 3.8.8.

### 4.1. VR-XGBoost Evaluation

#### 4.1.1. Construction of Feature Vector

We constructed the feature vector dataset for the training of VR-XGBoost by using 10 statistical features, as shown in Table 4. In the feature vector dataset, each vector contains 10 dimensions of attributes and its classification label *l*, where *l* = 0 represents non-VeESA and *l* = 1 represents VeESA. It is worth noting that the cumulative travel time $\gamma_{1}$ is not directly available in the ETC data. We replaced it with the cumulative travel time from the entrance of the toll station to the front gantry of ESA, i.e., $\gamma_{1}$ is the cumulative travel time from the entrance of the entrance to $G_{2}$.

The correlation heatmap is further inscribed for correlation analysis of the feature vectors, as shown in Figure 5. In the figure, blue indicates a positive correlation between vectors, and red indicates a negative correlation between feature vectors. At the same time, when the color is more prominent, the correlation between vectors is stronger. The speed features were positively correlated with the traffic flow and negatively correlated with the cumulative travel time $\gamma_{1}$, vehicle type $\theta_{1}$ and entrance gross axle weight $\theta_{2}$. Specifically, there is a strong positive correlation among $v_{1},\;v_{3}$ and $v_{4}$, which are both weakly positively correlated with $v_{2}$. The two features of $\gamma_{2}\;\mathrm{and}\;\gamma_{3}$ have a very low correlation with other features. Through heatmap analysis, we can clearly understand the correlation between feature vectors.

#### 4.1.2. Parameters Selection

The XGBoost classification algorithm has numerous parameters, including the following three aspects.

(1)General Parameters: booster, silent, nthread, etc.(2)Booster Parameters: the number of decision trees (n_estimators), learning rate (learn_rate), maximum depth of the tree (max_depth), minimum weight in leaf nodes (min_child_weight), parameter that controls the number of leaves (gamma), proportion of sample sampling (subsample), scale of feature sampling (colsample_bytree), etc.(3)Learning Task Parameters: objective and evaluative (eval_metric).

The general parameters and learning task parameters are set directly according to the model needs, while the booster parameters should be parameter-seeking by the tuning method. At present, the tuning method is mainly the grid search method, which is combined with the K-fold cross-validation method to achieve the optimal parameters [42]. In this work, we also used this method for tuning the parameters and set the cross-validation parameter K = 5. The search range, step size and optimal values of parameters for each parameter are shown in Table 5.

#### 4.1.3. Comparative Analysis of Classification Models

To comprehensively evaluate the effectiveness of the VR-XGBoost model, this work introduced evaluation metrics such as accuracy, precision, recall, and F1-score, as shown below.
(32)$$\mathrm{Accuracy}=\frac{TP+TN}{TP+FP+FN+TN}$$
(33)$$\mathrm{Precision}=\frac{TP}{TP+FP}$$
(34)$$\mathrm{Recall}=\frac{TP}{TP+FN}$$
(35)$$F1-\mathrm{score}=\frac{2\times Recall\times Precision}{Recall+Precision}\;$$

We compared and analyzed this experimental model with commonly used machine learning models (e.g., RF, GBDT, KNN), and the experimental results are shown in Table 6. The experimental results showed that VR-XGBoost, RF and GBDT all obtained good recognition results with accuracy above 95%, while DT performed the worst due to its tendency to overfit. In particular, VR-XGBoost achieved the best results in evaluation metrics. Specifically, in Part B, the accuracy of VR-XGBoost was as high as 97.4%. This result showed the significant superiority of the VR-XGBoost model for the recognition of VeESA.

Next, the feature contributions are further analyzed, as shown in Figure 6. As a whole, the feature contribution ranking from largest to smallest is speed features, external features, and spatiotemporal features. In particular, the contribution of the speed feature in Part A and Part B, both of which exceed 65%, is much higher than that of the spatiotemporal feature and external features. Specifically, the feature contribution of $v_{2}$ in speed features is more than 50%, indicating that the feature is the most important. In contrast, the contribution rates of features, such as the actual cumulative travel time $\gamma_{1}$, the time period feature $\gamma_{2}$, the passenger/freight volume $\theta_{2}$, and the traffic flow $\theta_{3}$, etc., are all less than 5%. These features seem less important. Through quantitative analysis of contribution rate, we can clearly know the importance of each feature.

### 4.2. K-VDTE Evaluation

We sliced in 5-min increments to count the dwell time, and the distribution is shown in Figure 7. It can be seen that the dwell times in Part A and Part B both exhibit a long-tailed distribution, which indicates that most vehicles stay in the ESA only temporarily and briefly. Specifically, the number of vehicles with a dwell time of 5~10 min is the greatest, and more than 90% of the vehicles have a dwell time of less than 1 h in the ESA. Furthermore, the average dwell time in Part A and Part B was approximately 30 min, with a standard deviation of approximately 70 min, a minimum dwell time of less than 30 s and a maximum dwell time of more than 12 h. Through the statistical analysis, we can clearly understand the general situation of the vehicle dwell time in each ESA.

To evaluate the effectiveness of the K-VDTE model, the estimation errors are quantified using the evaluation metrics of root mean square error (*RMSE*), mean absolute error (*MAE*), and *R* coefficient, as shown below:(36)$$RMSE=\sqrt{\frac{1}{n}\overset{n}{\underset{i=1}{\sum }}{\left(y_{i}-{\hat{y}}_{i}\right)}^{2}}\;$$
(37)$$MAE=\frac{1}{n}\overset{n}{\underset{i=1}{\sum }}\left|y_{i}-{\hat{y}}_{i}\right|$$
(38)$$R^{2}=1-\frac{\sum_{i=1}^{n}{\left(y_{i}-{\hat{y}}_{i}\right)}^{2}}{\sum_{i=1}^{n}{\left(y_{i}-{\bar{y}}_{i}\right)}^{2}}\;$$
where ${\hat{y}}_{i}$ denotes the estimated dwell time obtained using the model, $y_{i}$ denotes the true dwell time, ${\bar{y}}_{i}$ is the average dwell time, and *n* denotes the amount of data.

We compared the proposed K-VDTE model with the traditional averaging speed method and commonly used machine learning models, as shown in Table 7. The experimental results show that the proposed K-VDTE method performs the best, while the machine learning model performs the worst. Specifically, taking Part B as an example, the *MAE* of the K-VDTE model was only 14 s, which was not only more than one times better than the average speed method but also at least four times better than the machine learning models. This demonstrated the higher accuracy of our method. Moreover, comparing the *RMSE* of each model, the proposed K-VDTE model improved at least one order of magnitude over the machine learning models, which indicated that the proposed method was more robust.

Moreover, the integrated learning models, such as XGBoost, RF and GBDT, among machine learning models, perform better on evaluation metrics, while the single models, such as Lasso and KNN, obtain very poor results on all evaluation metrics. This result indicates that single models may not be suitable for vehicle dwell time estimation.

To investigate the estimated errors in depth, we performed a statistical analysis of the *MAE* of the dwell times and carved out the distribution of the cumulative probabilities, as shown in Figure 8. It can be seen that the distribution curves of the cumulative probabilities in Part A and Part B all exhibited a rapid increase with the increase in the dwell time estimation error until they stabilized after 2 min. Specifically, P{MAE≤120 s}>95% indicates that the probability of keeping the *MAE* within 2 min is more than 95%. Specifically, taking Part B as an example, the probability of controlling the *MAE* within 1 min and 2 min are $P_{B}\left\{MAE\leq 60\;s\right\}>97\%\;\mathrm{and}\;P_{B}\left\{MAE\leq 120\;s\right\}>99.8\%$, respectively. The results further validate that the K-VDTE model has strong robustness.

## 5. Conclusions

In this work, we proposed a method for the recognition of vehicles entering expressway service areas and the estimation of dwell time based on ETC data. This method provides reference ideas for scientific and reasonable evaluation of the service capacity of the ESA, which can also provide decision support for the optimization of the layout when reconstructing and extending the ESA. The specific conclusions are as follows:(1)Experiments were conducted using real ETC data with a user penetration rate of over 80%. It not only solves the issue of insufficient data volume but also solves the geographical differences existing in different service areas in vehicle dwell time estimation. It can provide a more scientific and reasonable reference basis for the evaluation of the service capacity of ESA.(2)Considering multidimensional information such as speed features, spatiotemporal features and external features, we constructed a VR-XGBoost model. This model can achieve not only the estimation of the overall pause rate of ESA but also the accurate recognition of vehicles entering the service area.(3)After an in-depth study of the driving pattern of vehicles in the process of driving in/out of the ESA, we proposed a K-VDTE to realize vehicle dwell time estimation. The estimation accuracy of vehicle dwell time can be further improved by considering vehicle kinematics.

However, the present method also has certain limitations, whose expressway traffic state must approximate free-flow conditions. In the future, we will further explore the vehicle driving characteristics and laws under nonfree flow conditions to form a more scientific and reliable evaluation system.

## Figures and Tables

**Figure 1 entropy-24-01208-f001:**
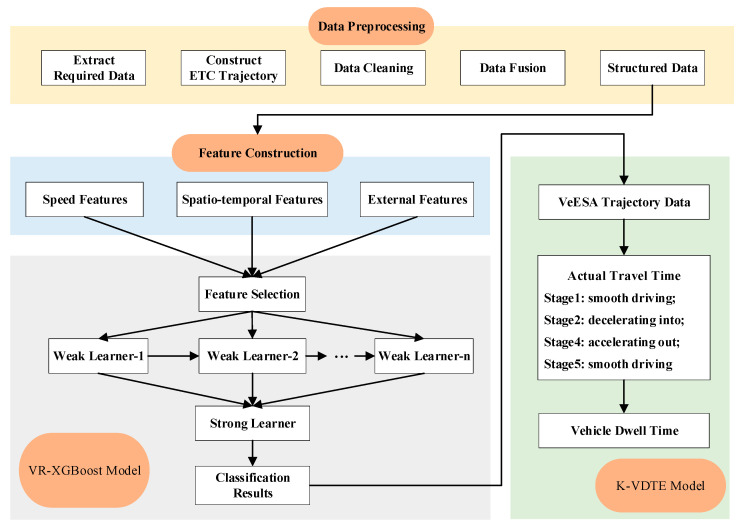
Overall framework.

**Figure 2 entropy-24-01208-f002:**
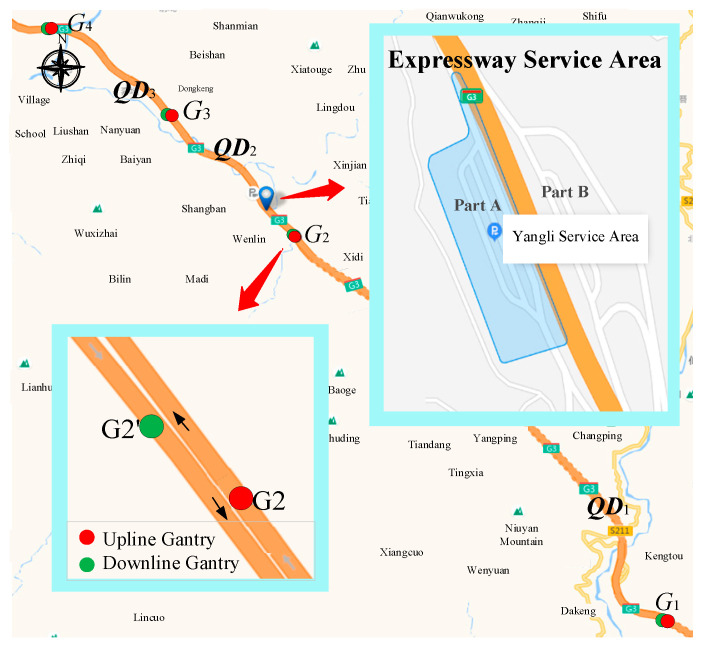
Visualization of ETC gantries and ESA locations.

**Figure 3 entropy-24-01208-f003:**
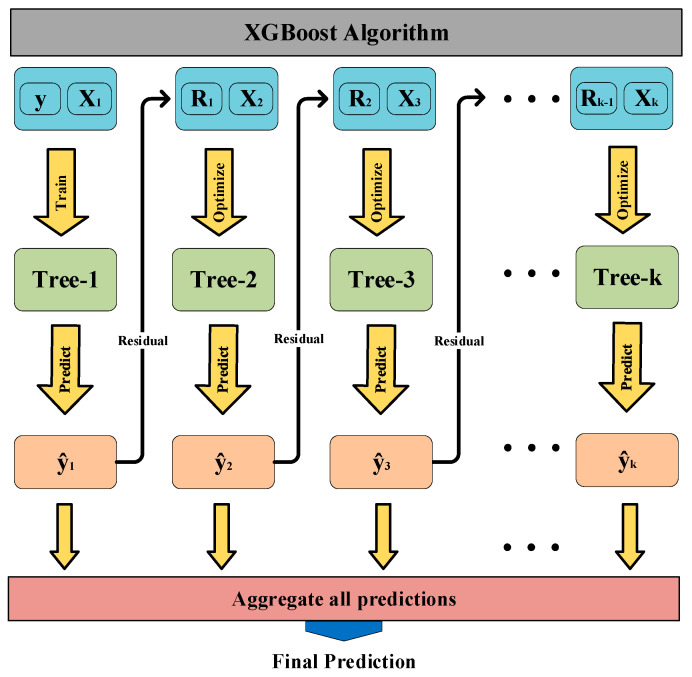
XGBoost Schematic.

**Figure 4 entropy-24-01208-f004:**
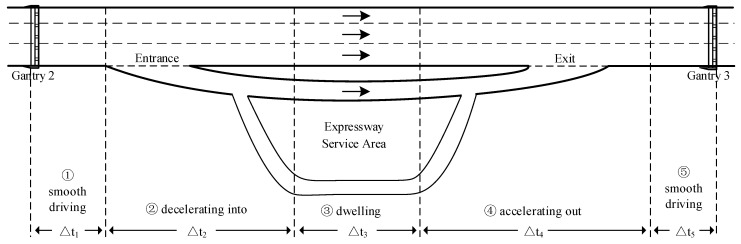
Kinematic process of vehicles driving in and out of the service area.

**Figure 5 entropy-24-01208-f005:**
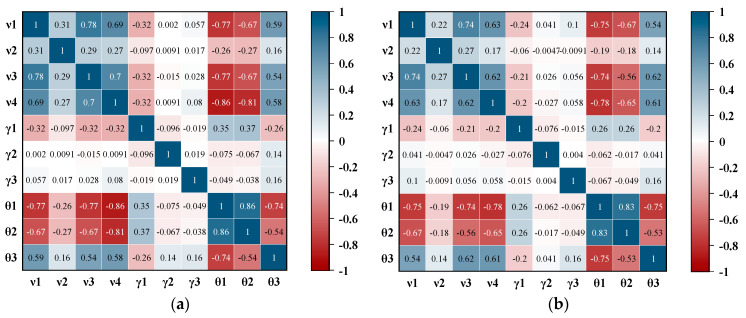
Correlation analysis of feature vectors. (**a**) Part A; (**b**) Part B.

**Figure 6 entropy-24-01208-f006:**
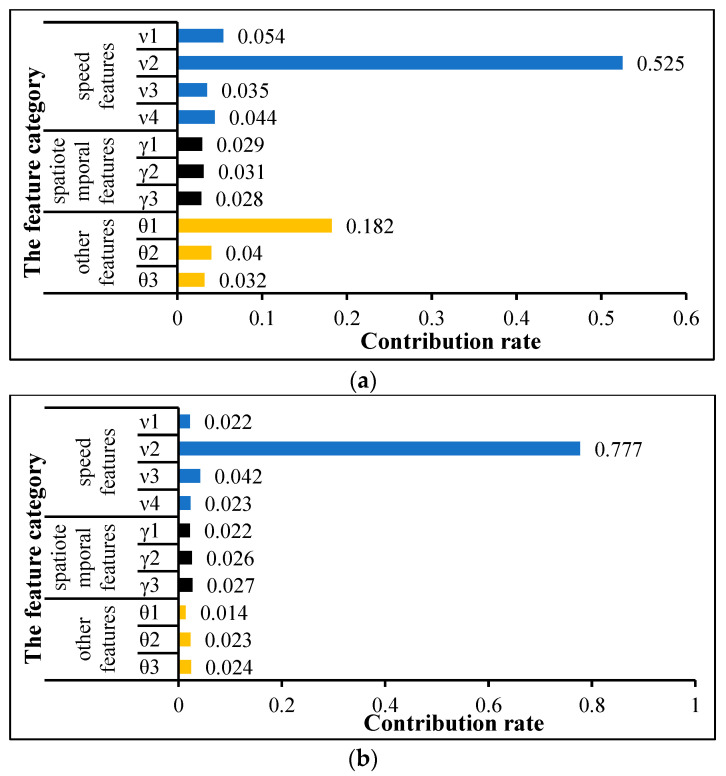
Contribution rate of features. (**a**) Part A; (**b**) Part B.

**Figure 7 entropy-24-01208-f007:**
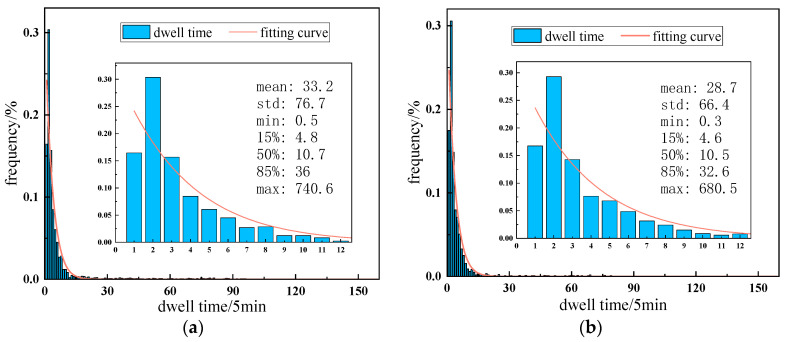
Dwell time distribution. (**a**) Part A; (**b**) Part B.

**Figure 8 entropy-24-01208-f008:**
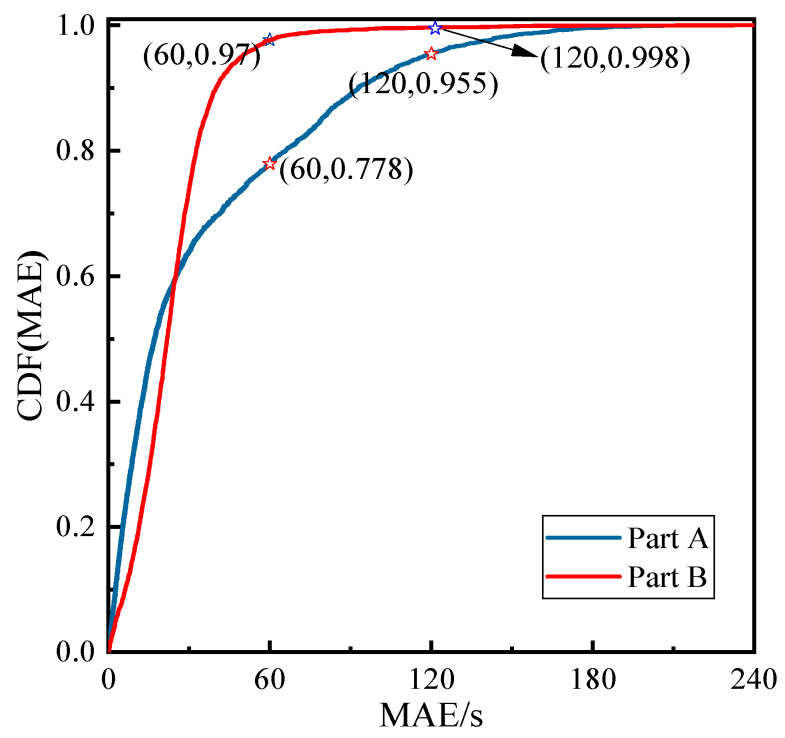
Cumulative probability distribution of *MAE*.

**Table 1 entropy-24-01208-t001:** Description of partial fields in ETC data.

	Field Name	Description	Example
1	VehID	vehicle ID	A000001
2	VehClass	vehicle type	1
3	EnWeight	entrance gross axle weight	1500
4	EnStation	entrance ID	1002
5	EnTime	entrance time	2020/9/5 00:00:00
6	GantryID	gantry ID	G000335001000120020
7	TradeTime	transaction time	2020/9/5 01:00:00
8	Workday	workday	0

**Table 2 entropy-24-01208-t002:** Description of fields in ESA data.

	Field Name	Description	Example
1	SAID	service area ID	Yangli Part A
2	EnEx	entrance/exit	0/1
3	VehID	vehicle ID	A000001
4	CapTime	capture time	2020/9/5 00:00:00

**Table 3 entropy-24-01208-t003:** Examples of experimental data.

	$tr_{p}$	$tr_{1.T}$	$tr_{2.T}$	$tr_{3.T}$	$tr_{4.T}$	$tr_{C}$	$tr_{EID}$	$tr_{ET}$	$tr_{W}$	$tr_{H}$	$tr_{PCT}$	$tr_{NCT}$	$tr_{l}$
Part A	A0000001	2020-09-05 08:06:03	2020-09-05 08:08:20	2020-09-05 08:14:12	2020-09-05 08:23:02	23	6101	2020-09-05 06:29:55	18.8	1	2020-09-05 08:01:59	2020-09-05 08:04:08	1
	A0000002	2020-09-03 06:28:34	2020-09-03 06:30:46	2020-09-03 06:43:52	2020-09-03 06:52:07	22	6103	2020-09-03 04:24:28	11.4	0	2020-09-03 06:24:42	2020-09-03 06:33:32	1
	A0000003	2020-09-10 23:38:27	2020-09-10 23:40:24	2020-09-10 23:43:03	2020-09-10 23:50:57	1	2202	2020-09-10 23:19:23	0	0			0
	A0000004	2020-09-07 03:51:13	2020-09-07 03:54:27	2020-09-07 03:59:52	2020-09-07 04:11:46	11	6101	2020-09-06 22:46:11	14.3	0			0
	A0000005	2020-09-03 21:14:13	2020-09-03 21:17:24	2020-09-04 04:56:05	2020-09-04 05:06:41	16	6307	2020-09-03 19:33:43	45.1	0	2020-09-03 21:12:24		1
Part B	A0000006	2020-09-04 17:17:52	2020-09-04 17:32:36	2020-09-04 17:48:00	2020-09-04 17:50:17	16	6707	2020-09-04 16:48:35	50.1	0		2020-09-04 17:36:41	1
	A0000007	2020-09-08 13:42:53	2020-09-08 13:54:00	2020-09-08 14:05:48	2020-09-08 14:07:57	2	6707	2020-09-08 13:23:20	0	0			0
	A0000008	2020-09-06 10:47:19	2020-09-06 10:55:17	2020-09-06 11:19:16	2020-09-06 11:21:21	3	2903	2020-09-06 09:52:21	0	1	2020-09-06 10:47:21	2020-09-06 11:08:32	1
	A0000009	2020-09-06 16:58:22	2020-09-06 17:07:12	2020-09-06 17:09:52	2020-09-06 17:12:13	12	6707	2020-09-06 16:37:20	7.6	1			0
	A0000010	2020-09-10 21:51:28	2020-09-10 22:01:59	2020-09-10 22:21:59	2020-09-10 22:24:20	14	6707	2020-09-10 21:25:11	17.9	0	2020-09-10 21:53:26	2020-09-10 22:09:52	1

**Table 4 entropy-24-01208-t004:** Sample of ESA feature vectors.

	$v_{1}$	$v_{2}$	$v_{3}$	$v_{4}$	$\gamma_{1}$	$\gamma_{2}$	$\gamma_{3}$	$\theta_{1}$	$\theta_{2}$	$\theta_{3}$	l
Part A	114.6	21.4	109.2	85.7	0.88	14	0	2	0	4	1
	92.0	93.4	84.9	92.1	1.01	10	0	2	0	3	0
	68.0	7.2	66.6	54.1	12.18	21	1	13	13.54	10	1
	75.4	69.6	69.3	48.6	1.86	21	0	14	15.9	11	0
	77.4	60.5	64.8	44.9	18.55	22	1	15	30.28	8	0
	70.0	64.4	77.8	68.4	2.72	15	0	21	0	4	0
Part B	67.6	21.2	79.6	84.1	0.81	17	0	12	9.3	6	1
	80.4	88.3	81.3	83.8	0.73	18	1	12	7.5	8	0
	77.0	20.1	86.1	74.4	0.56	20	0	11	4.6	16	1
	67.1	76.2	72.2	66.3	0.81	21	0	11	0	22	0
	90.1	9.7	104.7	94.9	0.42	22	1	1	0	69	1
	91.6	102.5	99.3	96.7	2.35	23	0	1	0	20	0

Notes: $v_{1}\sim v_{4}$: km/h; $\gamma_{1}$:h; $\gamma_{2}$: o’clock; $\theta_{2}$: t; $\theta_{3}$: veh; $\gamma_{3}$, $\theta_{1},$l: dimensionless.

**Table 5 entropy-24-01208-t005:** Optimal combination of parameters.

	Parameter	Search Range	Step Size	Optimal Value
General Parameters	booster	gbtree/gblinear		gbtree
	silent	0/1		0
	nthread			4
Booster Parameters	n_estimators	[100, 1000]	100	300
	learn_rate	[0, 0.5]	0.01	0.1
	max_depth	[1, 10]	1	5
	min_child_weight	[1, 10]	1	1
	gamma	[0, 0.5]	0.1	0
	subsample	[0.6, 1]	0.05	0.8
	colsample_bytree	[0.6, 1]	0.05	0.8
Learning Task Parameters	objective	reg:linear/reg:logistic/ binary:logistic/…		binary:logistic
	eval_metric	error/auc/rmse/…		auc

**Table 6 entropy-24-01208-t006:** Performance comparison of classification models.

Model	Part A				Part B			
	Accuracy	Precision	Recall	F1-Score	Accuracy	Precision	Recall	F1-Score
GaussianNB	0.937	0.94	0.937	0.937	0.962	0.962	0.962	0.962
SVM	0.954	0.954	0.954	0.954	0.973	0.974	0.973	0.973
KNN	0.955	0.956	0.955	0.955	0.973	0.974	0.973	0.973
DT	0.913	0.914	0.914	0.914	0.947	0.947	0.947	0.947
AdaBoost	0.941	0.942	0.941	0.941	0.969	0.97	0.969	0.969
LR	0.947	0.947	0.947	0.947	0.966	0.966	0.966	0.966
RF	0.958	0.96	0.958	0.958	0.973	0.974	0.973	0.973
GBDT	0.958	0.959	0.958	0.958	0.973	0.974	0.973	0.973
**VR-XGBoost**	**0.959**	**0.96**	**0.959**	**0.959**	**0.974**	**0.974**	**0.974**	**0.974**

**Table 7 entropy-24-01208-t007:** Performance comparison of estimation models (unit: s).

Model	Part A			Part B		
	*RMSE*	*MAE*	$R^{2}$	*RMSE*	*MAE*	$R^{2}$
Lasso	4046	2095	0.275	3831	1823	0.2
KNN	3443	1148	0.475	3506	1073	0.33
AdaBoost	536	431	0.987	486	400	0.987
DT	318	90	0.995	263	65	0.996
ExtraTree	365	92	0.994	1248	146	0.915
RF	276	71	0.997	263	55	0.996
GBDT	272	72	0.997	315	61	0.994
XGBoost	242	70	0.997	263	62	0.996
AvgSpeed	85	71	1.000	36	30	1.000
**K-VDTE**	**69**	**52**	**1.000**	**22**	**14**	**1.000**

## Data Availability

Restrictions apply to the availability of these data. Data were obtained from Fujian Expressway Information Technology Co., Ltd. and are available from the authors with the permission of Fujian Expressway Information Technology Co., Ltd.

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
