# Peer review of "Recognition of Vehicles Entering Expressway Service Areas and Estimation of Dwell Time Using ETC Data"

_entropy, 2022, doi:10.3390/e24091208_

Round 1
Reviewer 1 Report (New Reviewer)
The authors propose a method for recognition of vehicles entering expressway service areas and estimation of dwell time using ETC data to scientifically and effectively evaluate the service capacity and provide a decision-making supporting basis for the reconstruction and expansion of the expressway service area. This work has important theoretical significance and practical application value. At the same time, the experimental results verify the effectiveness of this work. The following issues are my concerns.
1. In Subsection 3.2, the authors should explain what is “Gantry 1 and Gantry 2 constitute Section 1 (QD1)” . The same are “Gantry 2 and Gantry 3 constitute Section 2 (QD2)” and “Gantry 3 and Gantry 4 constitute Section 3 (QD3)”
2. In Algorithm 1, I think the authors should explain “what is the topology data TP” and “what is the opposite topology data TP’”
3. In Section 4, more elaboration on results can have a better impact on the work.
Author Response
Dear Reviewer,
Thank you for your valuable suggestions on our submitted manuscript.
We have responded point-by-point to your suggestion as follows:
Point 1: In Subsection 3.2, the authors should explain what is “Gantry 1 and Gantry 2 constitute Section 1 (QD1)” . The same are “Gantry 2 and Gantry 3 constitute Section 2 (QD2)” and “Gantry 3 and Gantry 4 constitute Section 3 (QD3)”
Response 1: For the convenience of understanding, we give the definition of the Section in Subsection 3.2.1, as follows:
Expressway Section QD: Each ETC gantry and entrance/exit of an expressway toll station is collectively called a node G, and two adjacent nodes constitute an expressway section, referred to as QD:
| QD = <G1,G2> |
(1) |
where G1 and G2 are the start and end points of QD.
At the same time, QD1、QD2 and QD3 are visualized in Figure 2. Taking road upline as an example, it can be seen from Figure 2 that G1 and G2 constitute Section 1 (QD1), G2 and G3 constitute Section 2 (QD2) where the ESA is located, and G3 and G4 constitute Section 3 (QD3). It can be found from the partial enlarged detail that the gantries always appear in pairs and are distributed upline and downline of the road, such as <G1,G2> and <G1,G'2>.
Point 2: In Algorithm 1, I think the authors should explain “what is the topology data TP” and “what is the opposite topology data TP’”.
Response 2: We have explained TP and TP' in the corresponding sections of the manuscript as follows:
Topology dataset includes two subsets: TP and TP', which is a collection of topologies (e.g. <G1,G2>). Specifically, TP ={<G1,G2>,<G2,G3>,<G3,G4>,...}denotes normal topology data and TP'={<G1,G'2>,<G2,G'3>,<G3,G'4>,...} denotes opposite topology data. The topologies in and always appear in pairs, such as and <G1,G2> and and <G1,G'2>.
Point 3: In Section 4, more elaboration on results can have a better impact on the work.
Response 3: We agree that the recommendations made by the reviewer are pertinent. Therefore, We have reorganized and in-depth analysis of various parts of the experimental results. In particular, the feature correlation analysis in Section 4.1.1, the feature contribution rate in Section 4.1.2, and the statistical analysis of dwell time in Section 4.2 are all deeply analyzed and summarized. Please refer to the revised manuscript for the remaining specific details.
Thank you again for your hard work.

Reviewer 2 Report (New Reviewer)
This paper describes a method for recogniciton of vehicles entering expressway service areas and estimaation of the dwell time using ETC data.
Although the other abbreviations are explained at their first occurence, the ETC is not at all. This should be corrected.
The method is well described and seems to be sound.
The structure of the paper is relatively good, only third-level headings in Section 4.2 could and should be avoided.
The figures are appropriate and of good quality.
The references in the paper seem to be relevant and up to date.
The English is relatively good, but there are some typos and errors. Proofreading by a grammar-skilled native speaker is encouraged.
Author Response
Dear Reviewer,
Thank you for your valuable suggestions on our submitted manuscript.
We have responded point-by-point to your suggestion as follows:
Point 1: Although the other abbreviations are explained at their first occurence, the ETC is not at all. This should be corrected.
Response 1: ETC is an abbreviation for electronic toll collection. We have made additions and corrections in the Abstract of the manuscript. At the same time, we have also carried out a comprehensive inspection of the abbreviations of the full manuscript, and we have alse made some additions and corrections, such as Internet of Vehicles (IoV), radio frequency identification (RFID), on-board unit (OBU) to and road side unit (RSU).
Point 2: The structure of the paper is relatively good, only third-level headings in Section 4.2 could and should be avoided.
Response 2: We agree that the recommendations made by the reviewer are pertinent. Therefore, we have removed the third-level headings in Section 4.2 to make the article more structured.
Point 3: The English is relatively good, but there are some typos and errors. Proofreading by a grammar-skilled native speaker is encouraged.
Response 3: We have revised the full text of the manuscript, including typos, errors, grammar, etc. For example, In line 60, “…locality where the ESA is located…” has been modified to ”regionality”. In line 228, “nature” has been modified to “characteristic”. In line 616, “MAE” is missing a letter "E" in line 616, which has been revised and completed. Please refer to the revised manuscript for the remaining specific details.
Thank you again for your hard work.

Round 2
Reviewer 1 Report (New Reviewer)
In this revised manuscript, I think the authors have answered my concerns. Hence, I recommend acceptance.
This manuscript is a resubmission of an earlier submission. The following is a list of the peer review reports and author responses from that submission.
Round 1
Reviewer 1 Report
The manuscript discusses how to derive exact position of Electronic Toll Collection Gantry from ETC transaction data and Vehicle trajectory data, and the experimental results show highly precision. It seems the manuscript gains a great progress in POI positioning, but I doubt its practice value, as it is not difficult to acquire ETC locations. As far as the positioning method is cornered, the followed points should be addressed:
1) why not use some clustering method to derive Gantry “center”? as there exist a lot of special vehicles and each will get a transaction position when passing ETC Gantry.
2) Figure 1 is not well demonstrative for the following definitions.
3) Form (4) should be stated in more detail, e.g., what is the meaning and structure of its cell?
4) How Form (10) is calculated, and why its cell value domain is {0,1, 2}?
5) 4.1.2 is still about data pre-processing, should be put in 4.1.
6) The difference of three types of transactions is note clearly stated, and how to distinguish missing transactions from correct transactions? Why the value of 2 claims a false transaction?
7) The sampling rate of trajectory data and its positioning accuracy should be given in the section of experimental evaluation.